# Assessing the Impact of Selected Attributes on Dwelling Prices Using Ordinary Least Squares Regression and Geographically Weighted Regression: A Case Study in Poznań, Poland

Cyprian Chwiałkowski [1],* , Adam Zydroń [1] and Dariusz Kayzer [2]

1. Department of Land Improvement, Environmental Development and Spatial Management, Poznań University of Life Sciences, 60-649 Poznań, Poland
2. Department of Mathematical and Statistical Methods, Poznań University of Life Sciences, Wojska Polskiego 28, 60-637 Poznań, Poland
* Correspondence: cyprian.chwialkowski@up.poznan.pl

**Abstract:** The price of dwellings is determined by a number of attributes among which location factors are usually the most important. Comprehensive analyses of the real estate market should take into account a broad spectrum of attributes including economic factors, physical, neighborhood and environment characteristics. The primary objective of the study was to answer the question of what determinants affect transaction prices within the housing market in Poznań. The analysis was performed on the basis of source data obtained from the Board of Geodesy and Urban Cadastre GEOPOZ in Poznań. In our study, we used two research regression methods: ordinary least squares and geographically weighted regression. The estimated models made it possible to formulate specific conclusions related to the identification of local determinants of housing prices in the Poznań housing market. The results of the study confirmed that the use of the proposed techniques makes it possible to identify attributes relevant to the local market, and, moreover, the use of spatial analysis leads to an increase in the quality of the description of the characteristics of the analyzed phenomenon. Finally, the results obtained indicate the diversity of the analyzed market and highlight its ambiguity and complexity.

**Keywords:** property market; hedonic price model; dwelling prices; regression analysis; locational indicators

## 1. Introduction

### 1.1. Market of Dwellings

The price of property depends on a number of factors, which, in principle, can be divided into physical, locational, environmental and economic characteristics [1–3]. A special role within the market of dwellings is played by locational factors, which usually rank highly in the hierarchy of market participants [4,5]. Location is an important attribute primarily because it is not possible to modify it, unlike most other attributes, e.g., the standard of finish can be upgraded by performing a major renovation. The way in which the location attributes interact with the characteristics of the immediate neighborhood especially in recent years has been one of the key elements of research related to the characteristics of the property market [6–9].

At the same time, it should be emphasized that for many potential buyers, other attributes such as the situation of the apartment on the floor can often be much more important than the location of the apartment. Market participants increasingly pay attention to mundane issues such as the insolation of the apartment. The assumption that only location affects prices is incorrect. If this concept were true then two exemplary apartment properties located in one building would reach an identical price during the transaction. Practice, on the other hand, proves that even when these apartments have the same area,

the transaction price can be divergent due to differences in other attributes. In addition, it should be remembered that one of the basic characteristics of a property regardless of type is heterogeneity, and as a result, among other things, attributes that are important for one type of property may be of marginal importance in the case of a property of a different type. Given these assumptions, comprehensive property market analyses should absolutely take into account the attributes belonging to each of the four defined groups. A review of the literature on the subject, carried out at an initial stage, made it possible to distinguish a number of important attributes within the property market belonging to each of the distinguished groups (Table 1).

**Table 1.** Attribute categories based on the literature.

| Category | Variable | Author |
|---|---|---|
| Physical characteristics of the apartment | Area of the apartment | Čeh et al. [10] |
| | Number of rooms | Escobedo et al. [11] |
| | Age of the building | Wu et al. [12] |
| | Type of housing unit | Park et al. [13] |
| | Availability of parking/garage | Ko et al. [14] |
| | Floor location | Szczepańska et al. [15] |
| | Building construction | Trojanek et al. [16] |
| | Form of ownership | Marano and Tajani [17] |
| | Basement availability | Ottensmann et al. [18] |
| | Standard of finishing | Tomal [19] |
| | Energy efficiency | Zancanella et al. [20] |
| Characteristics of neighborhood | Distance to city center | Payton et al. [21] |
| | Distance to schools | Sah et al. [22] |
| | Availability of public transport | Cordera et al. [23] |
| | Distance to stores | Heyman et al. [24] |
| | Distance to recreational areas | Tyrväinen [25] |
| | Nuisance neighborhoods (airport, etc.) | Kopsch [26] |
| | Distance to industrial zones | Wittowsky et al. [27] |
| Characteristics of environment | Distance to lakes | Sander and Haight [28] |
| | Distance to green areas, parks | Łaszkiewicz et al. [29] |
| | Distance to legally protected areas | Pearson et al. [30] |
| | Air pollution | Chen and Chen [31] |
| | Distance to other valuable natural areas (e.g., mountains) | Xiao et al. [32] |
| Characteristics of economic conditions | Volume of taxes | Munro and Tolley [33] |
| | Demographic structure | Lai [34] |
| | Employment opportunities | Ding et al. [35] |
| | Prospects of economic development | Perdomo [36] |
| | Value of income generated | Feng [37] |
| | Degree of development of enterprises | Md and Sheikh [38] |
| | Occurrence of planning barriers | Hussain et al. [39] |
| | Value of macroeconomic factors | Li and Chu [40] |

*1.2. Poland's Property Market and Macroeconomic Conditions*

The property market is one of the most important sectors of the economy, and the share of outlays on housing, understood by both maintenance of the existing stock and new investments, usually accounts for 8–20% of Poland's GDP [41]. As a result of this relationship, property market processes are inextricably linked to the common conditions prevailing in the country's economy. As a result, economic factors should be an integral and key component of research related to property management [42–45]. Taking this into account, the introductory part of the work consists of a brief characterization of the macroeconomic situation over the past few years in Poland.

As part of the initial characterization of the database in question, which is the foundation of this work (dwellings transactions in 2019–2022 concluded within the boundaries of the city of Poznań), it should be noted that over the analyzed period of time, the macroeconomic conditions directly affecting the property market underwent significant changes. A special role in this process was played by the Monetary Policy Council, which is the decision-making body of the National Bank of Poland, which from the beginning of 2020 took a number of measures to mitigate the negative economic effects of the coronavirus pandemic. In order to intensify the level of investment and increase consumption, the aforementioned body has repeatedly decided to reduce the interest rates of the National Bank of Poland [46,47]. As a result, at the end of May 2020, interest rates reached record low levels not seen in the 21st century (reference rate—0.10, deposit rate—0.00, lombard rate—0.50, rediscount rate for promissory notes—0.11, discount rate for promissory notes—0.12). The aforementioned indexes were maintained at the indicated levels until the beginning of October 2021, when, due to the problem of drastic price increases gaining importance, the Monetary Policy Council decided to raise interest rates repeatedly. As a result, at the end of the analyzed time period, interest rates reached the following values: reference rate—3.50, deposit rate—3.00, lombard rate—4.00, rediscount rate for promissory notes—3.55, discount rate for promissory notes—3.60 [48]. In the following months, these values continued to be increased by subsequent decisions of the decision-making body of the National Bank of Poland.

Decisions related to interest rates directly affect the situation in the property market (especially housing); an increase in NBP interest rates leads to an increase in both the interest rate on the loan itself and interest, which automatically leads to an increase in loan installments. As a consequence, the number of people/entities that can apply for a mortgage is clearly decreasing due to the tightening of criteria related to the granting of credit, i.e., creditworthiness [49–52]. A special social group that may suffer the consequences of the described phenomenon may be the younger generations, who, as a result of a significant tightening of monetary policy, may not be able to purchase their own housing [53]. This phenomenon can ultimately lead to a significant increase in the level of poverty; very often it is even one of the main factors determining it [54]. In addition, it should be taken into account that this phenomenon can be compounded by the increased cost of housing, defined, for example, by the cost of electric energy, the prices of which have clearly increased in Central and Western European countries due to the pandemic but also the ongoing war [55]. As a result, it is extremely difficult not only to buy an apartment, but also to maintain it. Very often, it becomes more profitable and attractive to rent an apartment instead of buying it. A certain solution in such a situation for the cited social group may be to increase the supply of housing by, for example, intensifying housing investment within the non-commercial third housing sector (THS), the primary purpose of which being to improve the living conditions of households with low or medium levels of financial wealth [56]. At the same time, it should also be borne in mind that direct government intervention in the housing market, defined, among other things, by the introduction of temporary tax breaks for buyers of houses/apartments, can have a negative effect. In spite of the fact that such actions usually in the first stage lead to the revival of the market by increasing demand and raising prices, as a consequence they can have a clear impact on reducing the welfare of societies [57]. Taking this into account, it should be borne in mind that any centrally planned government interventions should be well thought out and well planned.

Based on NBP reports, it should be emphasized that in the case of apartments, the share of transactions financed by credit has been gaining importance in recent years and has become more popular in relation to transactions financed by own funds. In extreme cases, the purchase of as much as 68% (Q2 2020) and 50% (Q3 2021) of apartments in Poland was financed with credit [58]. As a consequence of the tightening of the criteria for granting mortgages, the demand in the housing market may significantly decrease which, if the supply remains constant, may lead to a marked decrease in transaction prices [59,60].

On a national scale, additionally taking into account the processes associated with the tendency to overvalue property prices [61,62], after several years of price increases in Poland, there is a real possibility of a significant decline in housing prices, while taking into account the differences in the potential of regional markets related to local regulations and socio-economic conditions [63–66]. The property market, due to its characteristics, reacts to the events described above with some delay. Consequently, although their impact on the processes taking place within them is undeniable, their effects will be visible in the long term.

With reference to the conditions outlined above, this study attempts to answer the question of what types of attributes significantly affect the prices of residential real estate within the analyzed market. The primary objective of the study was to identify the factors that are most relevant to market participants at a time of changing macroeconomic conditions. The analysis carried out is further aimed at verifying whether there is spatial autocorrelation of transaction prices for the analyzed dataset, as well as comparing the built models in terms of accuracy in describing the studied phenomenon.

## 2. Materials and Methods

### 2.1. Study Site and Data

The study in question was carried out within the administrative boundaries of the city of Poznań, which has the status of capital of the Wielkopolska Voivodeship and plays a key role in the social and economic development of neighboring administrative units. The city is located in the western part of Poland, while within the Wielkopolska Voivodeship it is situated in its central part (latitude of the city—northern hemisphere, longitude of the city—eastern hemisphere). The total area of the city equals 262 km$^2$. Within the boundaries of Poznań there are 40 cadastral precincts (districts); the largest covers an area of 13.78 km$^2$ (Kobylepole), while by far the smallest is the Daszewice—0.06 km$^2$ (Figure 1).

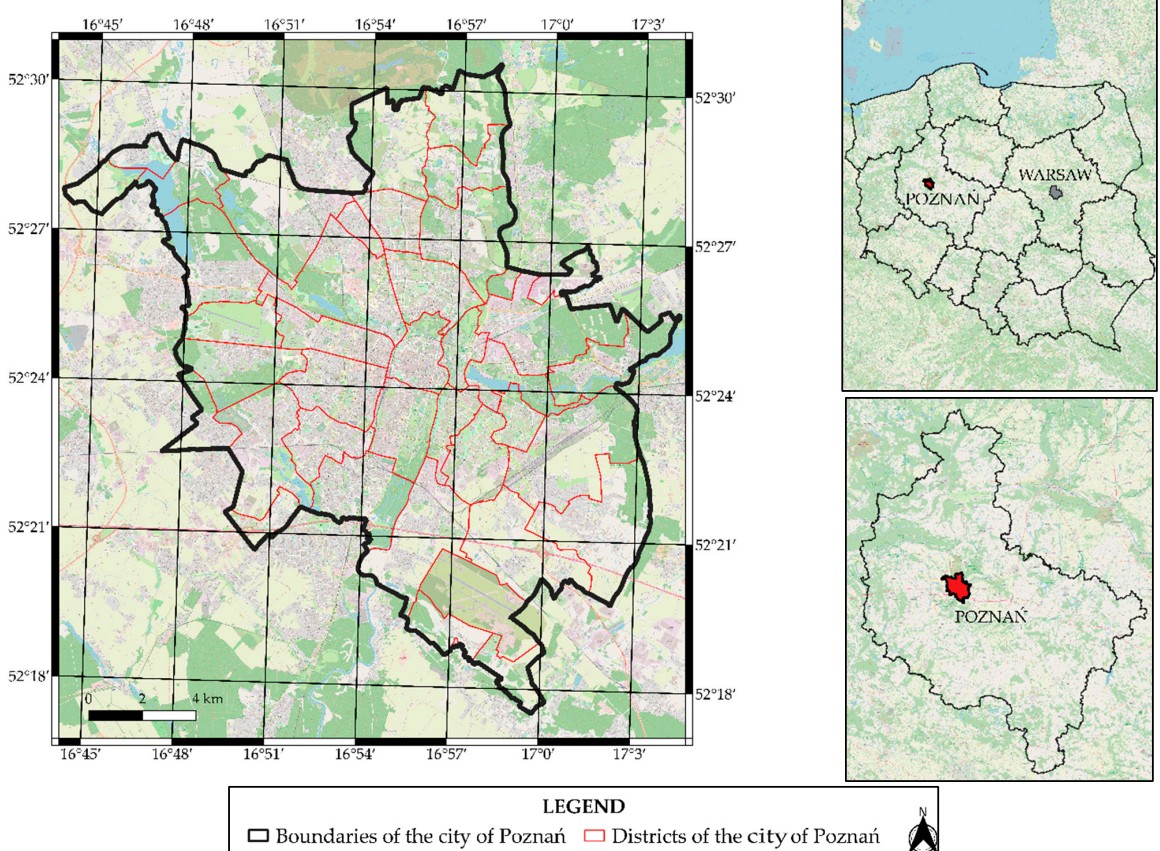

**Figure 1.** Location of the research facility—Poznań.

According to the Central Statistical Office, the city's population in 2021 was 529,410. A clear majority in the total population were women (about 53%, i.e., 282,307), while men made up a minority of the city's population (nearly 47%, i.e., 247,103). People who are of working age constitute a clear majority in the demographic structure of the population (nearly 60%); 25% are post-production age, and the remaining 15% of the city's population is made up of people of pre-production age (under 15). A detailed analysis of the state of the population over the past few years within Poznań makes it possible to note, first of all, a non-significant but fairly systematic reduction in the total population (2019—534,813 people, 2020—532,048 people) which may be partly related to the COVID-19 pandemic. According to the Central Statistical Office, more than 506,000 people died in Poland in 2021 (the most since World War II); in 2020, this figure was about 477,000 Poles which, compared to 2019 (409 thousand), demonstrates the extremely significant way in which demographics have been affected by both the pandemic itself and the not-always-effective measures taken to limit its spread.

The analysis underpinning this study was carried out on the basis of source data obtained from the Board of Geodesy and Urban Cadastre GEOPOZ in Poznań. The most relevant component of the data in question is the transactions of housing units that were concluded within the city boundaries of Poznań from the beginning of January 2019 to the beginning of March 2022. The key task of the analysis was to identify attributes that significantly affect prices in a changing economic environment. The selected dataset ultimately consisted of 6611 residential real estate transactions concluded in 2019–2022. The vast majority of transactions were concluded in the central part of the city (districts: Łazarz, Jeżyce and Old Town) and partially in the northern part of Poznań (districts: Piątkowo and Naramowice) (Figure 2).

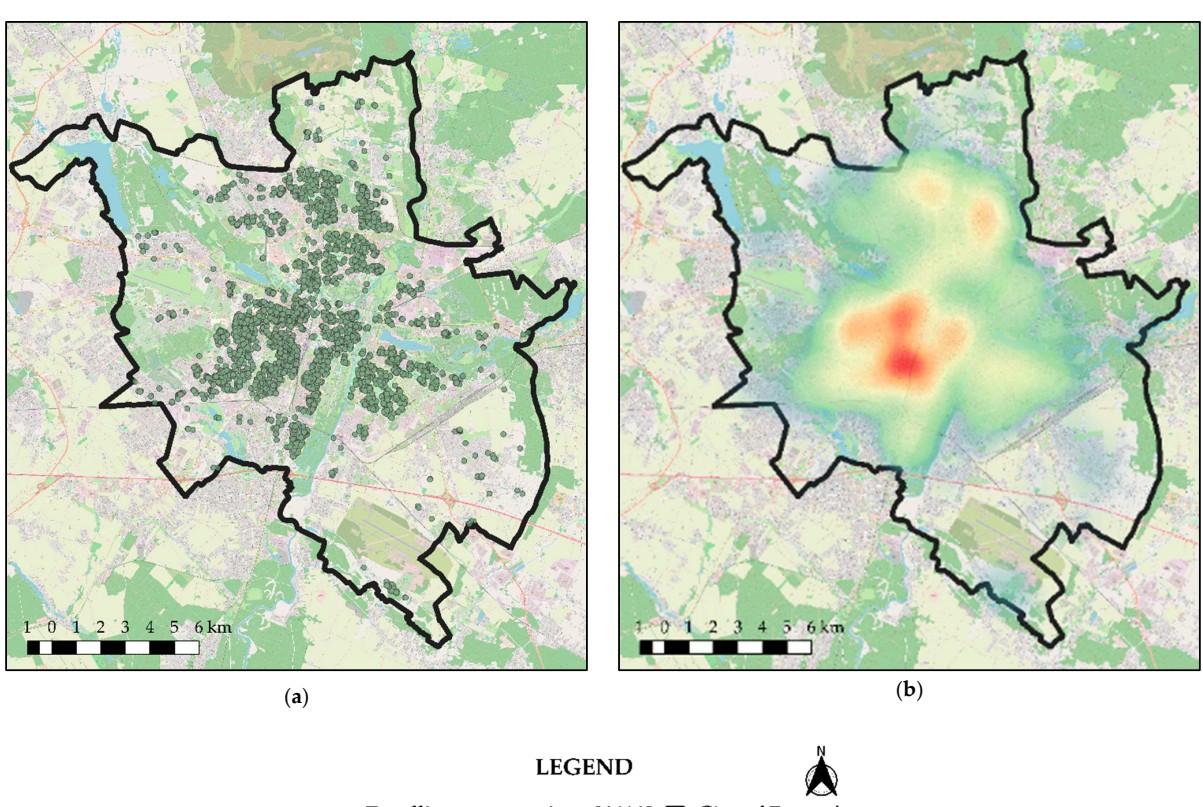

(**a**)　　　　　　　　　　　　　　　　　　　　　　　(**b**)

**LEGEND**

● Dwellings transactions [6611]　▣ City of Poznań

**Figure 2.** (**a**) Location of housing units analyzed. (**b**) Transaction density within the study area.

The basic data recorded in the subject database include the most relevant information related to each transaction, i.e., the transaction price, the date of the transaction, the area of the apartment, the area of auxiliary premises and information about the floor on which the

apartment is located in the building. The main advantage of this type of data is its source. Due to the fact that these data are derived from notarial deeds, they constitute a reliable and credible database, which is indispensable, among other things, during the process of monitoring the current situation in the real estate market and the characteristics of the mechanisms occurring within a given market. Despite the numerous advantages of this type of data, it is necessary to bear in mind certain problems associated primarily with the lack of complementarity in terms of the characteristics of the property according to the basic features that can directly determine the transaction price. An analysis that aims to identify attributes of key importance without taking into account characteristics such as, among others, the closest neighborhood, the technical condition of the building or the standard of finish (undefined in the surveyed register) may be unjustified and presumably may result in erroneous conclusions. In order to eliminate the above problem, the basic source data were additionally supplemented with features related to the locational characteristics of the analyzed properties.

The average unit prices of the collected transactions were in the range of PLN 3000 per sqm to PLN 16,000 per sqm in the extreme case. The lowest average prices were recorded in districts located far from the city center, such as Głuszyna, Spławie and Kotowo. On the other hand, the highest average prices, often exceeding PLN 10,000 per sqm, were recorded in the central part of the city in districts such as Jeżyce, Old Town and Wilda. Based on the collected transactions and their locations, a map of interpolated average prices in places where it was possible to determine them was prepared (Figure 3).

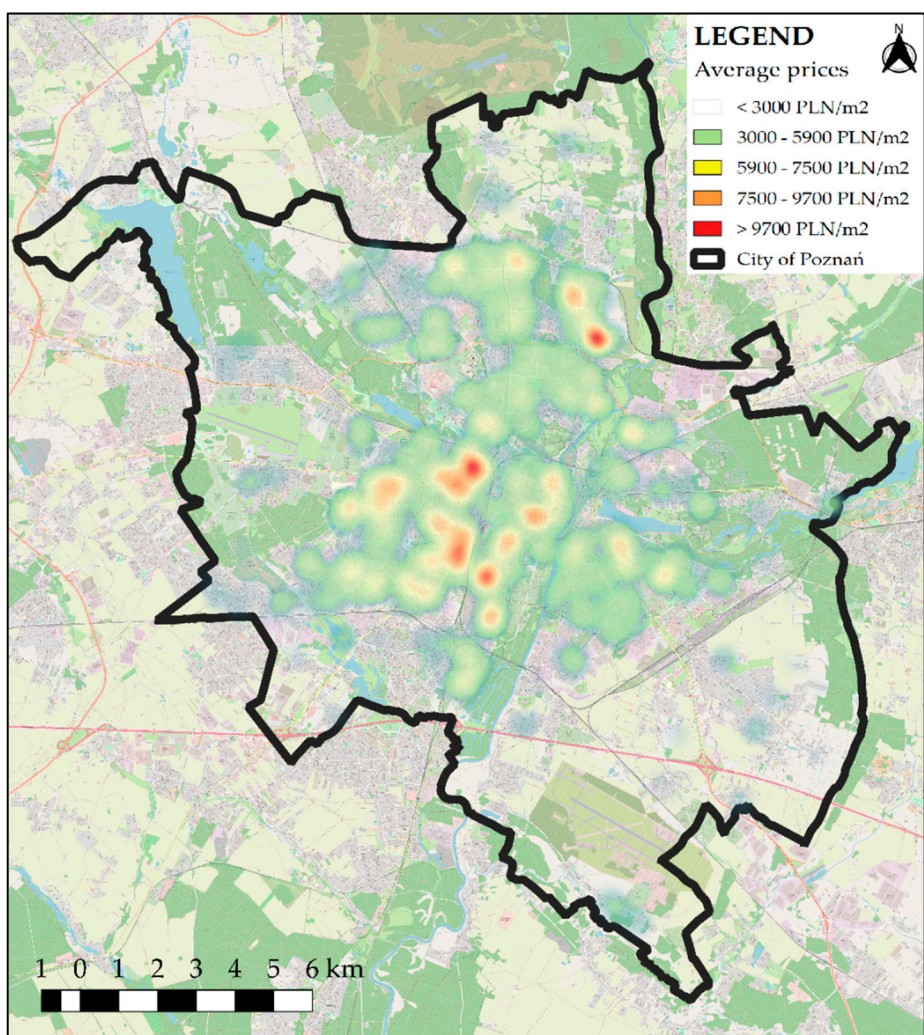

**Figure 3.** Interpolation of housing prices within the study area (kernel density estimation).

### 2.2. Regression Analysis

The hedonic regression method (HPM) is the foundation of research related to the process of analyzing how selected attributes affect the price of property, and the first studies related to it date back to the early 20th century. The first pioneering analysis performed using HPM involved the modeling of agricultural land prices [67]. Subsequently, the method served as a useful tool, among other things, in the process of characterizing the relationship occurring between the transaction price of dwellings and the level of air pollution [68]. In the second half of the 20th century, the theoretical framework of the HPM method was developed in a thorough and precise way by subsequent authors [69–71].

According to the basic assumption of the method in question, the price of property is determined by a number of attributes adopted into the model, a concept that is the fundamental essence of HPM models. Given this assumption, the key task of the estimated model is to answer the following question: to what extent do the independent variables included in the model affect the price of property? As a result of the applied statistical procedure, the obtained model allows for the defining of the values of the analyzed attributes, which are further components of the dependent variable, i.e., price. Finally, according to the above assumptions, the price of an apartment (or any other property) can be presented in the form of a standard regression equation:

$$P = f(LN, H, \beta, e) \tag{1}$$

where:

$P$—equal price of the apartment sold,
$LN$—a combination of locational and immediate neighborhood-related features,
$H$—a combination of features related to the physical characteristics of the apartment,
$\beta$—a combination of features related to the characteristics of local planning and social and economic conditions,
$e$—standard error.

After taking the appropriate independent variables into consideration, the estimated model of the dependent variable can be represented by a multivariate regression equation in the following form:

$$P = \beta_0 + \beta_1 \times X_1 + \ldots + \beta_n \times X_n + e \tag{2}$$

where:

$\beta_0, \beta_1 \ldots, \beta_n$—regression coefficients.
$X_1, \ldots, X_n$—values of the analyzed characteristic.

### 2.2.1. Ordinary Least Squares

The ordinary least squares (OLS) technique is one of the most widely used methods in regression analyses related to the definition of mechanisms occurring within the property market. OLS is a basic linear modeling method, and according to the fundamental theoretical assumptions of this method, the relationship between the dependent variable and the independent variables can be defined by a simple straight line for which the values of y (price) are estimated by the values of x (attributes). It is extremely important that in the predicted model the sum of squares of the errors of the estimated parameters is as small as possible [72]. Finally, the global regression model can be rearranged in the following form:

$$P = \beta_0 + \sum_{i=1}^{K} \beta_i X_i + e \tag{3}$$

where $P$ is the unit price of the apartment sold, logarithmic if the normality of the distribution is not met.

The basic problem of typical methods of statistical analysis, including the OLS technique, is related to the fact that, as a rule, these methods aim to formulate common relationships that exist between the variables under study in different locations [73]. How-

ever, it is problematic and impossible to achieve analyses that are representative of a given location, and the standard OLS regression may be insufficient to correctly identify the attributes that determine property sales prices. In addition, an extremely important issue is the fact that the findings obtained as a result of the OLS technique may be biased due to the heterogeneity of spatial relationships; moreover, in many cases the transactions collected and included in the analysis may be spatially autocorrelated. This phenomenon is directly related to the typical behavior of the average property market participant, who, when selling an apartment, takes into account the transaction prices of comparable housing units that were transacted in the immediate vicinity. The phenomenon of spatial autocorrelation may furthermore result from the fact that locational attributes can affect prices in certain areas in an analogous way [74]. In the case of the analyzed dataset, the occurrence of the phenomenon of spatial autocorrelation of the analyzed prices was checked using Moran's I test.

2.2.2. Geographically Weighted Regression

The solution to the problem arising from the phenomenon of spatial autocorrelation is the geographically weighted regression (GWR) method proposed in 1996 [75]. Within certain limits, the GWR technique can be defined as a kind of extension of the OLS global regression method, with the difference being that the GWR method allows the estimation of local coefficients based on samples within the bandwidth of a local location. Given the above considerations, the GWR model can be presented according to the following formula:

$$P = \beta_0\,(x_i, y_i) + \sum_{i=1}^{K} \beta_i(x_i, y_i)X_i + e \tag{4}$$

where:

$x_i$, $y_i$—$i$-th point geographical coordinates,
$\beta_0(x_i, y_i)$—location-specific intersection point,
$\beta_i(x_i, y_i)$—coefficient specific to the location of the point $i$,
$X_i$—variable related to $\beta_i(x_i, y_i)$,
$K$—number of estimated parameters,
$e$—standard error.

Parameters in the GWR model are estimated in a comparable way to classical techniques; however, an important difference is the assumption according to which weights are taken into account, being conditional on the location of individual observations:

$$\beta'(x_i,\ y_i) = \left(X^T W(x_i,y_i)X\right)^{-1} X^T W(x_i,\ y_i)y \tag{5}$$

where $W(x_i,y_i)$ is a diagonal matrix of weights, which are a function of the distance between the location given by the coordinates $(x_i, y_i)$ and the location of each point where the observation occurred.

The weights are usually determined using a function with a shape similar to that of a Gaussian curve, while taking into account the bandwidth of the parameters, which characterizes the spatial range from which the observations are taken into account in the calculations. The GWR results more accurately represent the global model as a result of using a larger bandwidth. As a result of applying the GWR model, a series of points defined by the estimated parameters is obtained, which makes it possible to observe the variation in these parameters in the analyzed area.

A direct comparison of the equations associated sequentially with the OLS model and the GWR model makes it possible to conclude that the global model can be interpreted as a kind of special case of the local model. The most significant difference between the compared models is the assumption that the parameters in the global model are taken as constant, while the coefficients in the model built based on GWR spatially vary. The key process during the implementation of the GWR regression is the calibration of the equation during which the assumption is made that observations occurring close

to the location of a given point have a significantly greater impact on the estimation of certain parameters compared to observations located at a greater distance. The coefficients of the model built using the GWR technique are estimated locally using the weighted least squares method, which consequently results in closer observations having a greater weight compared to observations located at a greater distance [76–79]. In summary, the main difference between the OLS technique compared to the GWR technique is that the parameters in the global model are constant, while in the GWR model the coefficients have location-dependent variability.

## 3. Results and Discussion

### 3.1. Characteristics of the Variables Included in the Analysis

As a preliminary step, a synthetic characterization of the dependent variable and independent variables (included in the analysis primarily on the basis of information from real estate offices) was performed, within which basic descriptive statistics were calculated, including the mean, standard deviation and maximum and minimum values (Table 2).

**Table 2.** Descriptive statistics of dependent and independent variables.

|  | Mean | σ | Max | Min | Median |
|---|---|---|---|---|---|
| Price (PLN/m²) | 7126.45 | 1689.48 | 16,000.00 | 3019.32 | 6969.70 |
| Number of rooms | 2.92 | 1.07 | 8.00 | 1.00 | 3.00 |
| Surface area (m²) | 53.28 | 22.20 | 237.08 | 10.56 | 48.80 |
| Floor | 2.94 | 2.51 | 17.00 | −1.00 | 2.00 |
| Associated premise | 0.37 | 0.59 | 9.00 | 0.00 | 0.00 |
| Distance to shopping center (m) | 918.83 | 769.30 | 7478.58 | 0.01 | 770.24 |
| Distance to tram stop (m) | 666.32 | 824.83 | 7141.70 | 13.76 | 368.67 |
| Distance to park(m) | 530.18 | 497.06 | 5539.09 | 0.15 | 438.09 |
| Distance to city center (m) | 3641.53 | 1751.48 | 11,328.34 | 149.43 | 3451.39 |
| Distance to bus stop (m) | 205.51 | 126.74 | 1434.71 | 11.93 | 181.05 |
| Distance to water (m) | 551.48 | 362.16 | 1711.63 | 18.64 | 461.84 |
| Distance to major roads (m) | 978.80 | 719.23 | 5340.19 | 9.23 | 846.90 |
| Distance to school (m) | 678.78 | 504.87 | 4387.43 | 2.92 | 572.47 |
| Number of observations |  |  | 6611 |  |  |

During the initial stage of the conducted analyses, the independent variables were subjected to an initial analysis. The issues of multilinearity between the adopted attributes and skewness defining the measure of asymmetry of the analyzed observations were taken into account in turn. In the study in question, attributes characterized by a skewness value greater than 3 were logarithmically transformed [80]. Then, the OLS regression was estimated with the use of the corresponding VIF (variance inflation factor) values which were simultaneously calculated for the analyzed attributes. Variables characterized by a VIF value greater than 7.5 were removed from the final part of the conducted analysis (Table 3).

### 3.2. OLS Regression

Based on the results obtained as a result of the OLS global regression model, it can be concluded that most of the variables are statistically significant when the *p*-value = 1%. The variables that are not statistically significant are F, DS and DPark. The obtained model explains about 23% of the variability of the observed phenomenon. Note that the negative values of the coefficients for factors determined by the measuring distance (e.g., for the trait distance from the center of Poznań—DCC) indicate a positive effect of the trait on the price of the properties included in the study. Variance inflation factor (VIF) coefficients for none of the analyzed attributes exceed the value of 3, which may indicate a lack of collinearity between the variables included in the analysis. Most of the variables received the expected

values of the coefficients; the exception may be the attribute related to the distance of the apartment from the tram stop. The negative impact of this attribute may be related to the noise generated by this mode of transportation (Table 4).

**Table 3.** Characteristics of qualitative and quantitative variables applied in the model.

| Feature | Symbol | Feature Description | Form |
|---|---|---|---|
| Price | P | Housing price (PLN/m$^2$) | Standard |
| Rooms | NR | Number of rooms | Standard |
| Surface area | SA | Surface area of the housing unit (m$^2$) | Standard |
| Floor | F | Floor number | Standard |
| Associated premise | BT | Number of associated premises | Standard |
| Shopping center | DS | Distance to the shopping center (m) | Logarithmic |
| Tram stop | DT | Distance to a tram stop (m) | Logarithmic |
| Park | DPark | Distance to a park (m) | Logarithmic |
| City center | DCC | Distance to the center of Poznań (m) | Standard |
| Bus stop | DB | Distance to a bus stop (m) | Standard |
| Water | DW | Distance to surface water (m) | Standard |
| Major roads | DMR | Distance to major roads (m) | Standard |
| School | DSch | Distance from educational institutions (m) | Standard |

**Table 4.** Summary of ordinary least squares (OLS) regression results.

| | Coefficient | Standard Error | T-Value | *p*-Value | VIF |
|---|---|---|---|---|---|
| Intercept | 9067.4456 | 211.9445 | 42.7822 | 0.0000 * | - |
| NR | −571.0740 | 26.2763 | −21.7334 | 0.0000 * | 2.3694 |
| SA | −3.6035 | 1.2695 | −2.8386 | 0.0046 * | 2.3928 |
| F | 12.7004 | 7.4552 | 1.7036 | 0.0885 | 1.0525 |
| BT | −180.8261 | 31.7043 | −5.7035 | 0.0000 * | 1.0616 |
| DS | −39.2856 | 52.0490 | −0.7548 | 0.4504 | 1.1284 |
| DT | 384.7857 | 60.1649 | 6.3955 | 0.0000 * | 2.0067 |
| DPark | −65.5485 | 45.8887 | −1.4284 | 0.1532 | 1.1633 |
| DCC | −0.1962 | 0.0173 | −11.3365 | 0.0000 * | 2.7684 |
| DB | 0.2517 | 0.1499 | 1.6791 | 0.0000 * | 1.0882 |
| DW | −0.2631 | 0.0601 | −4.3795 | 0.0000 * | 1.4262 |
| DMR | 0.1089 | 0.0313 | 3.4784 | 0.0005 * | 1.5280 |
| DSch | −0.1133 | 0.0408 | −2.7737 | 0.0056 * | 1.2811 |
| Number of Observations | | | | 6611 | |
| Multiple R-Squared | | | | 0.2330 | |
| Adjusted R-Squared | | | | 0.2316 | |
| AICc | | | | 17,035.3480 | |

\* An asterisk next to a number indicates a statistically significant *p*-value ($p < 0.01$).

In addition, the OLS regression analysis carried out made it possible to determine, on the basis of the model built, the estimated prices of the dwellings analyzed. The vast majority of the values of the standardized residuals indicating differences between the observed and estimated prices were in the −0.5 to 0.5 range (2868 observations). At the extremes, the values of standardized residuals amounted to −3.13 (the difference between the observed and estimated price was −4649.13 PLN) and 6.38 (the difference between the observed and estimated price was −9444.06 PLN) (Figure 4).

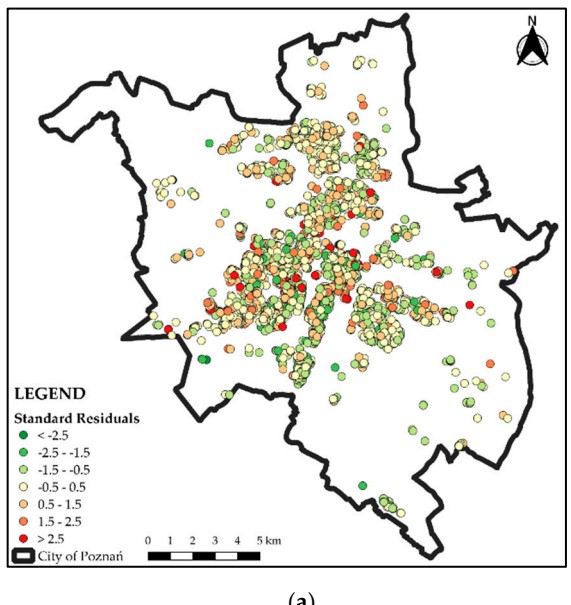

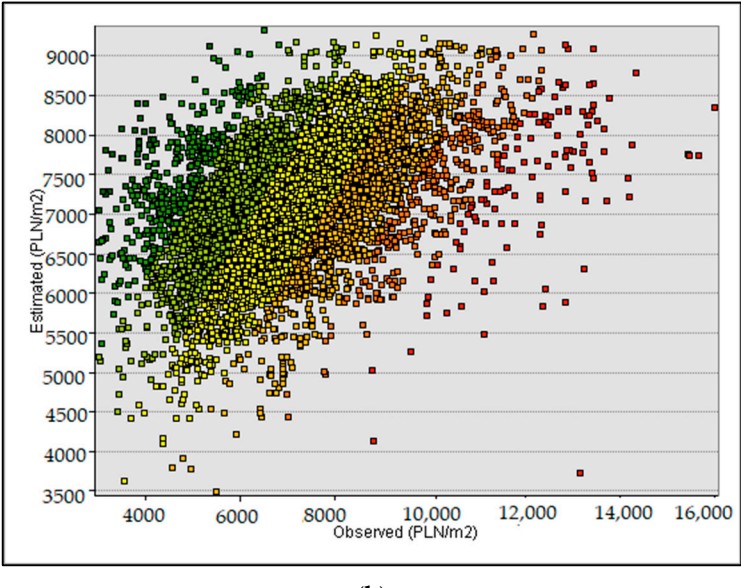

(**a**)   (**b**)

**Figure 4.** (**a**) OLS standard residuals. (**b**) Difference between the observed and estimated values.

### 3.3. GWR Regression

The GWR regression analysis was applied after first identifying the phenomenon of spatial autocorrelation of the analyzed prices; therefore, for the analyzed set of housing units, the value of Moran's I test was calculated. Based on the results obtained, defined by both the values of the test itself and the *p*-value, it can be concluded that the property prices are spatially clustered and are characterized by positive spatial autocorrelation, and the results obtained are statistically significant (Table 5).

**Table 5.** Measures of spatial autocorrelation for the dwelling prices.

| | |
|---|---|
| Moran's Index | 0.3904 |
| Expected Index | −0.0002 |
| Variance | 0.0001 |
| z-score | 54.7223 |
| *p*-value | 0.0000 |

For additionally enriching the analysis on the basis of the results obtained from the OLS regression, the phenomenon of spatial autocorrelation was checked using Moran's I test for OLS residuals. Considering the results including the z-score variance of 48.5599, there is less than 1% probability that the analyzed clustering pattern may be the result of random chance (Table 6).

**Table 6.** Measures of spatial autocorrelation for the OLS residuals.

| | |
|---|---|
| Moran's Index | 0.3464 |
| Expected Index | −0.0002 |
| Variance | 0.0001 |
| z-score | 48.5599 |
| *p*-value | 0.0000 |

MGWR 2.2 software was used to perform the geographically weighted regression [81]. The results obtained following GWR indicate a significant difference in the range of results obtained with respect to global regression (OLS). The value of the adjusted R-Squared is 0.477 (OLS: 0.2316), and the value of AICc is equal to 15597.432 (OLS: 17035.3480). Based on

the values presented thus far, we can conclude that the GWR technique has a more efficient modeling quality with respect to the OLS technique. Taking into account the specifics of this type of regression, the number of observations (defined by the percentage value) that are statistically significant based on the T-value (adj. critical t value is equal to 3.384) was further defined. The largest number of statistically significant observations was defined for the surface area feature (10.88%), while the smallest was defined for the distance to surface water feature (2.24%). Furthermore, the difference with OLS regression is the ability to analyze the phenomenon of collinearity of the analyzed attributes for each observation. The highest number of cases with a VIF value greater than 7.5 was identified for the DCC attribute (6178 observations); for the floor attribute all observations had a VIF value less than 7.5 (Table 7).

**Table 7.** Summary of squares (GWR) regression results (Group 1).

| | Mean | SD | Min | Median | Max | Percent of Significant Cases at 95% | Percent of Cases with Local VIF > 7.5 |
|---|---|---|---|---|---|---|---|
| Intercept | −6.208 | 57.630 | −451.945 | −0.312 | 503.614 | 3.43% | - |
| NR | −0.170 | 0.231 | −1.004 | −0.165 | 0.640 | 7.94% | 10.29% |
| SA | −0.235 | 0.309 | −1.310 | −0.239 | 0.942 | 10.88% | 9.08% |
| F | 0.064 | 0.154 | −0.392 | 0.039 | 0.830 | 3.84% | 0.00% |
| BT | −0.032 | 0.579 | −3.730 | −0.032 | 33.803 | 6.84% | 0.64% |
| DS | 0.015 | 8.460 | −72.226 | −0.012 | 126.927 | 2.87% | 73.21% |
| DT | 3.717 | 27.158 | −23.143 | 0.116 | 302.379 | 4.77% | 51.88% |
| DPark | −0.289 | 1.667 | −17.968 | 0.014 | 9.381 | 3.99% | 62.00% |
| DCC | 1.133 | 49.423 | −390.291 | 0.229 | 499.569 | 3.15% | 93.45% |
| DB | 0.116 | 0.804 | −5.035 | 0.012 | 10.587 | 2.44% | 19.63% |
| DW | −0.311 | 1.954 | −19.191 | −0.287 | 11.336 | 2.24% | 68.40% |
| DMR | −1.406 | 22.238 | −143.849 | −0.092 | 283.861 | 5.19% | 87.49% |
| DSch | −0.604 | 3.352 | −39.026 | −0.127 | 24.271 | 4.39% | 70.17% |
| Number of Observations | | | | | | 6611 | |
| Multiple R-Squared | | | | | | 0.549 | |
| Adjusted R-Squared | | | | | | 0.477 | |
| AIC | | | | | | 15,309.007 | |
| AICc | | | | | | 15,597.432 | |
| Bandwidth used | | | | | | 167.000 | |

Based on a preliminary analysis of the average values of the parameters of the GWR model, we can conclude that the nature of the impact of most attributes is comparable to the estimates of the OLS model (the difference is in the DS, DCC and DMR attributes). However, it should be emphasized how important the minimum and maximum values of the coefficients are, which indicate how significantly the impact of a given attribute can vary within the analyzed area. The results obtained confirm the heterogeneity and complexity of the market under study, and underscore the validity of including the exact geographic location of observations in the analysis.

The model obtained with the GWR regression revealed that the most significant determinant of unit transaction prices for the housing market in Poznań as a whole was the variable related to floor area (SA). The significance of this variable was confirmed for 719 observations (a total of 6611 observations were taken). The SA attribute has, as expected, a negative impact on the amount of the unit transaction price of apartments in almost the entire analyzed area. In general, the strongest negative impact of the analyzed determinant (FA) can be observed among transactions located in the southeastern part of the city. However, the negative correlation between the analyzed attribute and price decreases in the center and northeastern part of the city, where a positive impact of the

analyzed independent variable on the dependent variable can be observed in selected areas. This phenomenon is observed, among others, in the district of Poznań—Old Town, which is considered, among other things, to be due to its location which is prestigious. In the case studied, this fact can directly result in an increase in the motivation of buyers to pay a higher price for each additional square meter of apartment space. At the same time, however, it should be emphasized that in many cases the positive correlation is statistically insignificant, which directly results in the fact that the correlations described above cannot be considered as undeniably certain (Figure 5).

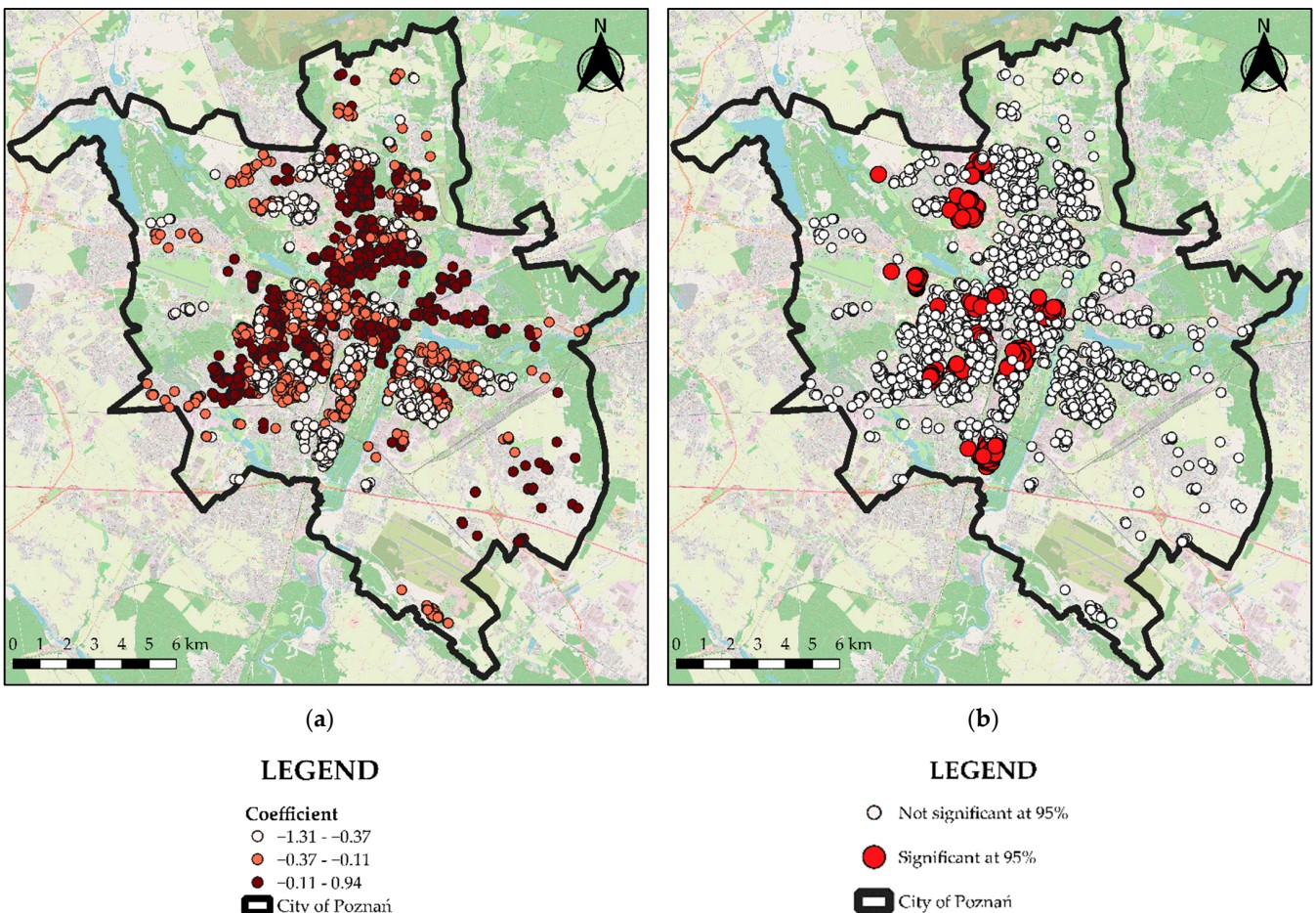

(**a**)　　　　　　　　　　　　　　　　　　　　　　　　　　(**b**)

**LEGEND**

**Coefficient**
○ −1.31 - −0.37
● −0.37 - −0.11
● −0.11 - 0.94
▭ City of Poznań

**LEGEND**

○ Not significant at 95%
● Significant at 95%
▭ City of Poznań

**Figure 5.** (**a**) Local estimated coefficients—SA. (**b**) Statistical significance of estimated coefficients—SA.

Among the locational characteristics, interesting results can be observed for the determinant related to distance from a bus stop. The number of statistically significant observations is small (161), while if the attribute is significant it is mostly in locations at a considerable distance from the city center.

This correlation may indicate that this attribute is particularly important for homebuyers, who in the future will have to use public transportation for an efficient connection to the center. For the aforementioned observations, in most situations the coefficient has a negative value whereby it can be concluded that in areas located on the outskirts of the city, the higher price is obtained by apartments located closer to the bus stop (Figure 6).

In comparison, the variable related to accessibility to a tram stop was statistically significant for 315 observations located in the city center, in most cases along the course of existing tram lines (Figure 7).

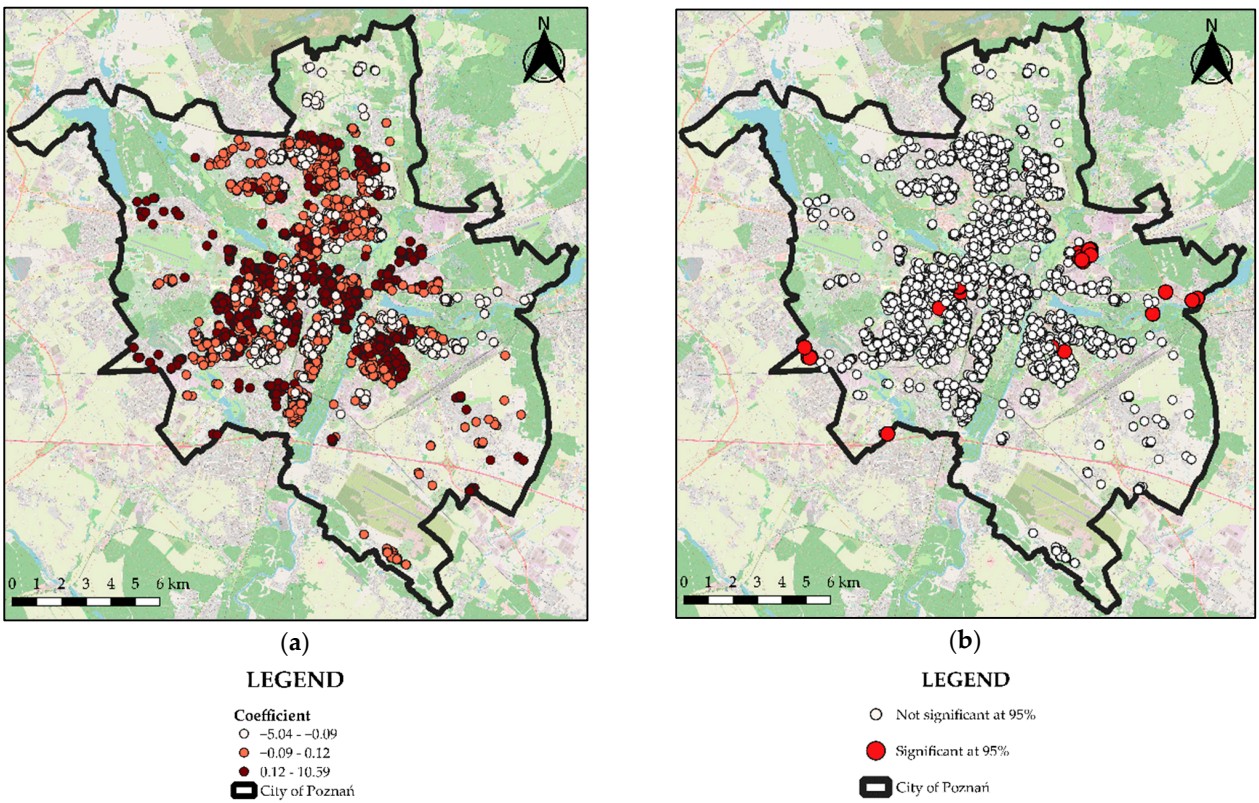

**Figure 6.** (**a**) Local estimated coefficients—DB. (**b**) Statistical significance of estimated coefficients—DB.

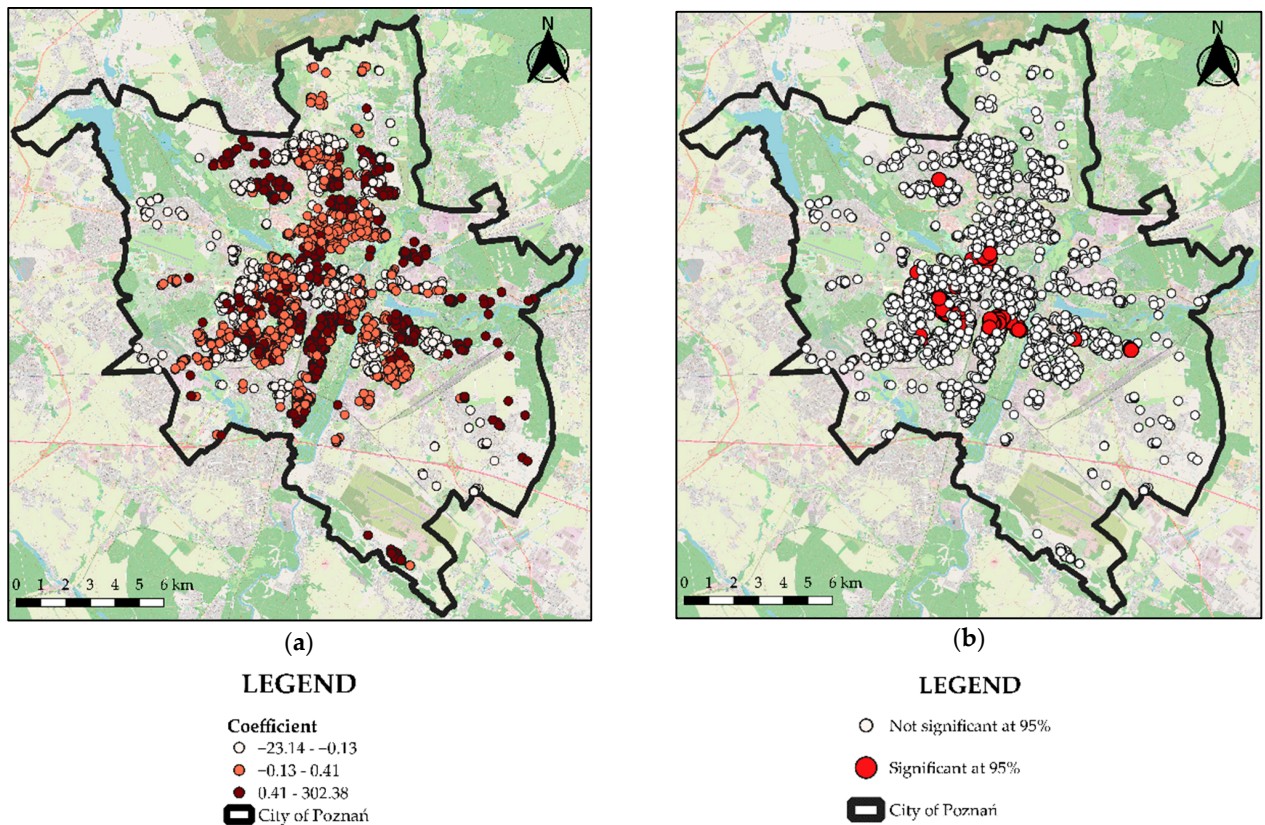

**Figure 7.** (**a**) Local estimated coefficients—DT. (**b**) Statistical significance of estimated coefficients—DT.

Due to the number of analyzed attributes and the complexity of the analyzed phenomenon, the remaining estimation results of the GWR model are presented in the Supplementary Materials (Figures S1–S9).

In order to systematize the analysis carried out and to present in a clear form the results obtained for the housing transaction market in Poznań, the number of significant attributes that determined the sales price is presented for each housing unit included in the analysis. The vast majority of transactions that were characterized by a higher number of relevant attributes (4–9) were located in the city center. This phenomenon may be due to the fact that the residential real estate market in this area is the most developed and most of the analyzed transactions were concluded within it (Figure 8). Moreover, it is worth noting that the results obtained confirm the phenomenon according to which certain variables can be consolidated into a single characteristic by market participants. An example is the characteristics related to the number of rooms and area of the apartment, which on a global scale have a significant impact on the level of transaction prices. On the local scale, however, in a small number of cases both price determinants are simultaneously significant.

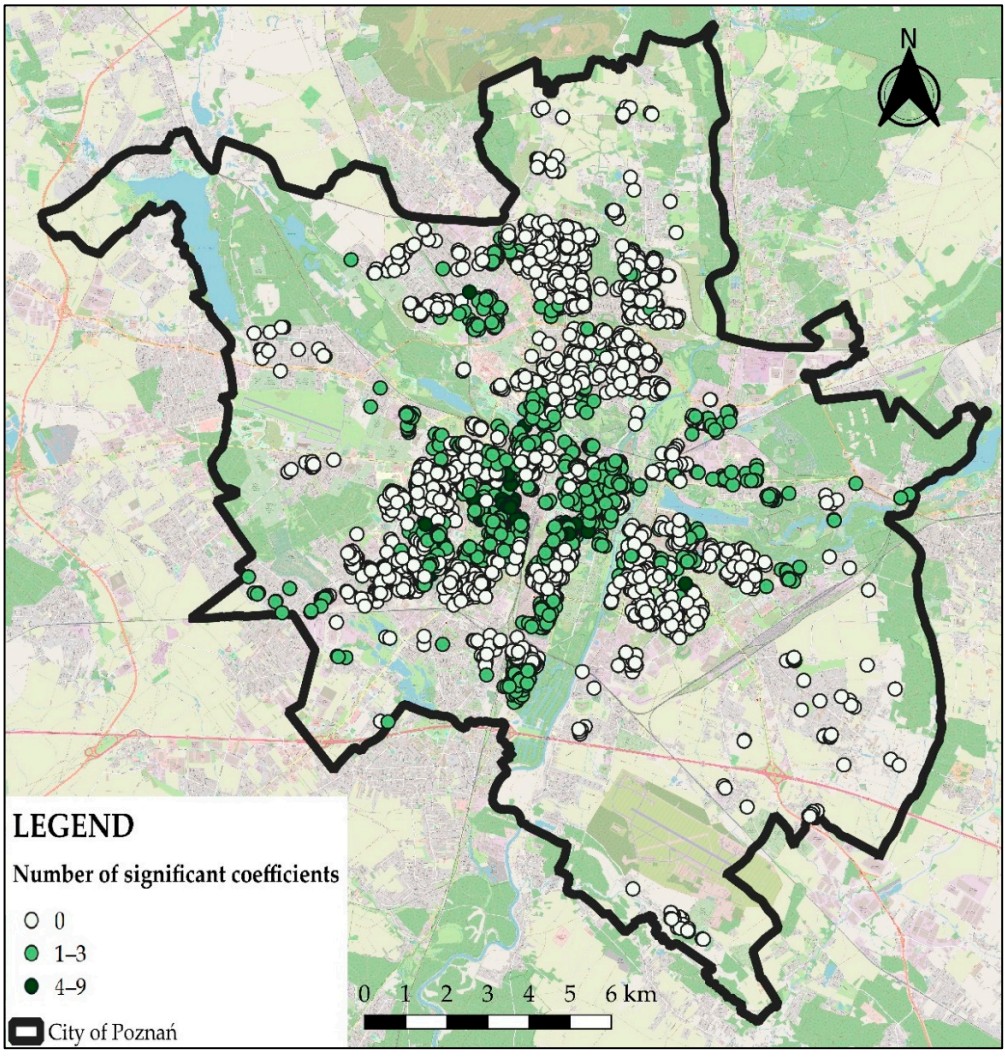

**Figure 8.** Number of significant coefficients for each dwelling at 95%.

## 4. Conclusions

The mechanisms taking place within the residential real estate market are essentially determined by two groups of factors: microeconomic and macroeconomic. The undisputed majority of existing scientific studies take into account in their structure a wide range of microeconomic features [82–85] or a single factor that is a fundamental element of the

analyses performed, which belongs to this group [86–90]. This study attempts to identify the determinants (belonging to the first group) that significantly influence prices within the housing market in Poznań. The factors belonging to the second group were only some kinds of external determinants, which, due to the need to present a detailed characterization of the market, were taken into account in Section 1.2 of this work.

An indisputable advantage of our study is its reliability, since the source data obtained and recorded came directly from the market. Analyses of this kind can be characterized by a much wider application compared to studies based, for example, on data derived from the bidding market or the rental market [91–93]. Additionally, with respect to some of the comparable studies conducted within selected Polish cities [94–97], the present analysis in its structure covers a relatively long period of time which also increases the applicability of our results.

The characteristics of localization issues in its structure should be defined by spatial autocorrelation analysis [98]. The key advantage of spatial regression models (e.g., GWR) compared to standard regression models (OLS) is that in their structure they take into account local spatial relationships, which allow results to be achieved that are characterized by a much greater degree of detail at the local level [99]. This regularity is confirmed by the GWR regression results we obtained, according to which the coefficients for each attribute included in the analysis are diverse and heterogeneous within the study area. The adoption of homogeneous conclusions as a result of OLS regression for the entire study area would constitute a serious error, which could consequently lead to fundamental cognitive errors. In addition, it is worth noting that the GWR analysis made it possible to identify (apart from local markets characterized by individual conditions) certain characteristics whose mode of influence is universal for the greater part of the city. Within the city of Poznań, these are variables related primarily to the physical characteristics of the apartment, i.e., the number of rooms and the area.

Despite the presented advantages of the methodology used in this study, it obviously has some limitations. First, gathering more precise information about the analyzed properties could make the analysis more complex. The use of other types of methods and the inclusion of additional characteristics related to, for example, the standard of housing units in the study could directly lead to a greater accuracy of the model built [100]. The process of selecting the appropriate attributes for analysis is also an undeniable problem. Although the variables were selected primarily on the basis of information obtained from local real estate offices, there can be no absolute guarantee that these are the most relevant attributes within the local market. There is a real possibility that an important attribute may have been overlooked and that the analysis includes an attribute that should not have been included.

**Supplementary Materials:** The following supporting information can be downloaded at https://www.mdpi.com/article/10.3390/land12010125/s1. Figure S1. (a) Local estimated coefficients—Number of rooms. (b) Statistical significance of estimated coefficients—Number of rooms. Figure S2. (a) Local estimated coefficients—Floor number. (b) Statistical significance of estimated coefficients—Floor number. Figure S3. (a) Local estimated coefficients—Number of associated premises. (b) Statistical significance of estimated coefficients—Number of associated premises. Figure S4. (a) Local estimated coefficients—Distance to the shopping center. (b) Statistical significance of estimated coefficients—Distance to the shopping center. Figure S5. (a) Local estimated coefficients—Distance to park. (b) Statistical significance of estimated coefficients—Distance to park. Figure S6. (a) Local estimated coefficients—Distance to the center of Poznań. (b) Statistical significance of estimated coefficients—Distance to the center of Poznań. Figure S7. (a) Local estimated coefficients—Distance to surface water. (b) Statistical significance of estimated coefficients—Dis-tance to surface water. Figure S8. (a) Local estimated coefficients—Distance to major roads. (b) Statistical significance of estimated coefficients—Dis-tance to major roads. Figure S9. (a) Local estimated coefficients—Distance from educational institutions. (b) Statistical significance of estimated coeffi-cients—Distance from educational institutions.

**Author Contributions:** Conceptualization, C.C. and A.Z.; methodology, C.C. and D.K.; software, C.C. and D.K.; validation, A.Z., C.C. and D.K.; formal analysis, C.C. and A.Z.; investigation, C.C.; resources, C.C.; data curation, C.C.; writing—original draft preparation, C.C.; writing—review and editing, A.Z. and D.K.; visualization, C.C.; supervision, A.Z.; project administration, C.C.; funding acquisition, C.C. and A.Z. All authors have read and agreed to the published version of the manuscript.

**Funding:** The publication was co-financed within the framework of the Ministry of Science and Higher Education program "Regional Initiative Excellence" in the years 2019–2022, project No. 005/RID/2018/19.

**Institutional Review Board Statement:** Not applicable.

**Informed Consent Statement:** Not applicable.

**Data Availability Statement:** The data are not publicly available due to restrictions on their use imposed by law and the entity collecting the data.

**Conflicts of Interest:** The authors declare no conflict of interest.

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
