# Peer review of "Assessing the Impact of Selected Attributes on Dwelling Prices Using Ordinary Least Squares Regression and Geographically Weighted Regression: A Case Study in Poznań, Poland"

_land, doi:10.3390/land12010125_

Round 1

Reviewer 1 Report (New Reviewer)

I have reviewed the paper title " Assessing the impact of selected attributes on dwelling prices using Geographically Weighted Regression: A Case Study in Poznań, Poland " and I have following reservations on it. 

first of all author mixed the files and not uploaded the accurate one. the 2nd thing author uploaded change track versions not the clean copy. 

The abstract is too short and not giving the concise overview of the research work.

Introduction part needs to be restructured and improvements required in term of latest literature work.

It would be a great idea if longitudinal/latitudinal value of the study area map is provided. 

The analysis carried out is very simple. the method section should be innovative. 

Insert grid to all the maps.

Conclusion should be precise. try to reduce it and be focused as much as you can. 

Author Response

Thank you very much for your valuable comments, which we have included in the revised version of our work. In the appendix we send the work with all the applied corrections (for 3 reviewers), in red color we marked comments directly related to your comments.

Reviewer 2 Report (New Reviewer)

Dear Authors,

congratulations for your interesting research. I have some suggestions for attracting more interest from international audience:

1: Your topic should be settled in a broader discussion on decreasing housing affordability accross countries. It has been found that young generation has minimum change to buy own housing not only because of increased housing prices (See here, for example:  DOI: 10.52950/ES.2021.10.2.003) but also because of increased costs and lower accessibility to mortgage financing (see here: DOI: 10.52950/ES.2022.11.1.007). Housing contributes to poverty nad level of deprivation and is an important topice ven in countries experiencing decrease in the overal poverty level (see here: DOI: 10.52950/ES.2022.11.1.009). I suggest referencing these examples from other countries. In this regards, non profit housing sector is an important concept in some countries (see here: DOI: 10.52950/ES.2022.11.2.002).

2:  It is advisable to reveal the importance of increased housing costs related to increased energy costs. In some countries this share of housing costs makes housing unaffordable even for households having their homes in own possession. This is a resonant issue in Europe currently (see here: https://doi.org/10.3390/en15041281) but perhaps in other countries. This aspect should be mentioned.

3: What I miss in this paper is a policy debate. Should the support (if any…) be aimed at supply or demand side of the market? See here: DOI: 10.52950/ES.2019.8.2.001

These are interesting points of view to be included and referenced in your research in order to attract more attention from international audience.

Good luck!

Author Response

Thank you very much for your valuable comments, which we have included in the revised version of our work. In the appendix we send the work with all the applied corrections (for 3 reviewers), in red color we marked comments directly related to your comments. 

Reviewer 3 Report (Previous Reviewer 3)

The authors have made a number of revisions, which make the paper more meaningful. However, the authors may still need to make some improvements:

1.     In the revised version the authors have revised the title of the paper, which emphasizes geographically weighted regression. However, OLS regression is also an important part of the authors' study.

2.     In 2.2. Regression Analysis, the reason that the author explained in the revised version that the two regression methods were not divided into different three levels of titles is not valid. The authors can refer to most of the current studies. In the model introduction, if the study adopts multiple regression model, it is better to use different titles. If the author does not accept this suggestion, please give other more sufficient reasons.

3.     The clarity of some figures in the paper is low.

I hope the author can seriously consider my comments and suggestions, in order to improve the content of the article.

Author Response

Thank you very much for your valuable comments, which we have included in the revised version of our work. In the appendix we send the work with all the applied corrections (for 3 reviewers), in red color we marked comments directly related to your comments. 

Round 2

Reviewer 1 Report (New Reviewer)

Thank you very much for revising the manuscript.

Reviewer 3 Report (Previous Reviewer 3)

The author has made a good revision according to the former comments, and I think that it could be published in the current form!

This manuscript is a resubmission of an earlier submission. The following is a list of the peer review reports and author responses from that submission.

Round 1

Reviewer 1 Report

General comments and grounds for negative opinion

The study asks whether the way in which the selected attributes characterising the analysed properties are affected by macroeconomic factors. The authors chose only one macroeconomic factor, i.e. the reference interest rates set by the National Bank of Poland.
The problem posed in the paper is rather strange, in any case atypical, as it concerns an attempt to find cause-and-effect relationships between quite distant areas. On the one hand, interest rates related to the fundamentals of the economy, and on the other, the way in which we perceive the characteristics of a property, such as its location on a floor or its distance from surface water. Indeed, hedonic models are that distinctive class of econometric models in which not the direct results of measurement, but the way in which the characteristics of a good are perceived, are the determinants of the dependent variable (price).
Is it really a scientific problem to answer the question whether a change in reference rates will, for example, make area less important and distance to green spaces more important? Of course, there should be no barriers in front of science but I think this trumps a banality. There is no justification as to why this scientific problem is worthy of attention. In my opinion, the problem is not scientific. It is too trivial. Of course, the influence of macroeconomic factors on the property market, understood not only as prices, is very important and always relevant. In this case, however, it is not about prices, demand or supply, but about the relationship between an event in the form of changes in interest rates (for the whole country) and the parameters of a hedonic model built on the basis of data from the local property market. Hence my negative opinion. I simply do not see the added value for science, from this kind of research.

The authors write that the primary objective of the study is to prove through empirical analysis the imperfect nature of the real estate market, manifested, inter alia, in the degree of its sensitivity to external factors. The fact that the real estate market is an imperfect market is one of the paradigms of the market and this has been confirmed in numerous publications. The reviewed publication could also testify to this but unfortunately does not. A perfect market, according to Adam Smith, is characterised by a high level of efficiency, which means reacting quickly to external factors. The property market reacts slowly, with a very long delay. The authors therefore contradict themselves. They claim that the property market is imperfect but reacts almost immediately to a change in macroeconomic conditions. I believe that this point has not been sufficiently considered by the authors.

The existing body of research on the causal relationship between macroeconomic factors and prices shows some divergence in results, which became particularly important after the US mortgage crisis more than a decade ago. However, most researchers argue that the effects of changes in macroeconomic factors (not just changes in interest rates) are observed in the real estate market with a rather long lag (a few months). Please note that the sale price is not fixed on the day of the transaction. In the case of mortgage financing, it can be set even six months earlier. In this study, the authors assume that changes in the market (indeed, changes concerning the parameters of the model) occur almost immediately after a change in interest rates. This constitutes, in my opinion, a clear error. There is also no reflection on the impact of the level of change and only the date of the change is noted.

The analysis carried out also aims to provide irrefutable evidence that the preferences of propertymarket participants change over time, as a result of which we have no way to make assumptions that will be universal and appropriate for markets in different locations and distinct time periods.
Of course, the preferences of market participants change over time. And this is an issue of real scientific interest. However, hedonic models do not provide conclusive evidence of preferences but only indicate them indirectly. The authors rightly conclude that it is not possible to make assumptions that will be appropriate for every local market, in every time period. This is, of course, an issue described in detail in the literature and there is no need to prove it once again.

Specific comments - for information only

Line 24-44
The excerpt contains trivial statements that may or may not, of course, be found in a scientific article.

Table 1.
It looks as if each author (or team of authors) is only concerned with one category of factors. In fact, the opposite is true.

Line 151-161
The dates underlying the groupings are conventional. According to the available data, there were no sharp changes in interest rates. Hence, the reaction of the property market can be expected to be strongly delayed.

Line 196-254
There is no need to explain the basis of the multiple regression model. A general explanation that a classical additive model was used is sufficient.

Table 2
Variables that only take integer values should not be presented as a decimal fraction (eg Number of rooms)

Table 3
There is no information on how the value of the Floor variable was determined. Is it assumed that the number of floor corresponds to the value of the variable? This does not correspond to the nature of the scales adopted in hedonic models.
The table shows that a logarithmic transformation of the variables DS, DT and DPark was performed. Why were the DS and DT variables not transformed in group 2. After all, the use of different scales automatically precludes full comparability of the models.

Line 313-314
"Prices may be determined primarily by economic factors, rather than by the locational or physical attributes of the housing units analysed". In my view, this is an opinion that is, to put it mildly, unauthorised. The economic factors obviously shape the overall price level, whereas the physical attributes only cause price differentiation within the local market area. For example, if the average price of 1m in Poznań is $2,000, the average is influenced by supply and demand (shaped by demographic and macroeconomic factors), while differences in prices of individual properties are due to differences in attributes.

Figure 3
There is no comment or reflection on what purpose Canonical Variate Analysis is intended to serve. It is certainly not a justification, in my opinion, to carry out the analysis separately in the three groups. That the structure of properties varies is natural. Some dwellings may be new and sold by developers. It is known that similar flats are then sold in series. As new developments are delivered, the characteristics of the flats sold change.

OLS regression
Note the VIF - it is considered a good indicator of colinearity but unfortunately it is not perfect. For example, it will not show a high correlation between the number of rooms and the area of the flat. I am convinced that there is collinearity here, which limits the reliability of the results.
The research did not include the date of the transaction as an independent variable. The available information (Eurostat) shows that there were significant changes in the price index in Poland during the analysed period. Not including such an important variable could have distorted the results of the study.

GWR regression
Spatial studies of the real estate market generally take into account spatial autocorrelation and spatial heterogeneity. In the former case, spatial autocorrelation models are used, while GWR models are used in the latter case. Therefore, the study of spatial autocorrelation (Moran I) in order to study spatial heterogeneity misses the point (in any case, it does no harm).
Presenting only general GWR statistics does not increase our knowledge of the phenomenon under study. As a rule, GWR results are presented on a map as the heterogeneity of the relationship between structural parameters and the dependent variable.

Reviewer 2 Report

General feedback:

The paper aims to identified how selected attributes characterizing the flats sold in Poznan between 2019 – 2022 are affected by macroeconomic factors. The macroeconomic factors are defined in the framework of this study by the level of interest rates. Authors added the dimension of time by dividing the pool of data into three groups. Although I know studies that also includes grouping to hedonic pricing method, I have never seen any study that includes time in the same way as here. This is highly innovative, but I am afraid authors did not manage to answer their research question. They also admit it in the discussion: “The most important element of the work was to test whether the changing macroeconomic environment directly affects the way the selected factors affect the price of the property. Such an approach, due to the difficulty of directly linking macro and micro factors, has not been of comprehensive interest to researchers, and our analysis can provide a reference point in the discussion related to the way macroeconomic factors affect local property market conditions”.

I believe that this study needs more consideration. It is a very general assumption that the macroeconomic factors can only be represented by interest rates. The interest rates are also problematic. Authors also admit that the real estate market responses with a time lag. Authors analyse only three years 01.2019 – 03.2022 which are years during which the real estate markets and general economy in the country has been affected by external changes. I believe to really be able to see the impacts, one should analyse at least three years before when the market was not affected by external impacts.

I believe that aiming at grasping the change in preference in the property markets over time is very important. However, for this reason the analysis should be reconstructed and probably redone with a different set of data. There is a lot of good work in the current state of the manuscript, but it is too confusing for the reader now.

Specific feedback:

·         Title is not clear. It is hard to understand what the paper is exactly about.

·         Abstract: The sentence with objective of the study is not very clear, there seem to be some syntax issue.

·         Abtract: The results obtained confirm the imperfection of the market, which is significantly dependent on external conditions – It seems that authors are using the wrong terms. Real estate market is not an isolated market that is not affected by huge external factors such a pandemic or increase in inflation. This does not mean that the market is imperfect. Imperfect markets are characterised by high barriers to entry and exit, there is no full disclosure of information about products and prices, and that individual buyers and sellers can influence prices and production.

·         Please describe how you decide on the elimination of the transactions from the initial pool? In this stage, you may significantly impact your final results: “The verification of the collected data (elimination of transactions, among others, whose unit prices clearly differed from average market prices)”

·         Figure 2: The d,e,f, figures are not really conveying much more information that a,b,c, I would rather see the hot spot analysis of the price per m2 to see how distance decay function from the city centre is performing in Poznan.

·         Why some distance variables enter the analysis in logarithmic form while others not and this also varies among groups?

Reviewer 3 Report

In this article, the author put forward a meaningful research perspective, and used the hedonic price model and the Geographically Weighted Regression method to conduct a rigorous empirical test. The results obtained confirm the imperfection of the market, which is significantly dependent on external conditions. Also, the research results provide a theoretical reference for the development of the real estate market in other regions. However, the article still has some problems that need to be revised before it can be published in land.

1.      In introduction, the author analyzed the current situation of the housing market and Poland's property market and macroeconomic conditions. However, there are many literatures on comprehensive factors that affect the housing market. If these literatures are considered in introduction, it will enrich the introduction.

2.      The macro policy changes in this study are divided according to interest rates. The increase or decrease range of interest rates may also be factors affecting the housing market. Does this study also consider this key factor? In addition, the author could add follow-up research work and research direction in the discussion. If there are specific interest rate data, please supplement the interest rate change chart in 2.1. Study Site and Data.

3.      In the section of 2.3. Region Analysis, the author could consider adding the titles of 2.3.1 Hedonic region method (HPM) and 2.3.2 GWR. A more detailed division would make the structure of the article more reasonable.

4.      As there are many tables in this article, please consider merging Table 4, Table 5 and Table 6 to one table, and retain the important characteristics of the three sets of regression equations. At the same time, some descriptive statistical tables can exist in the form of attached tables.

5.      In this article, the author demonstrated mature empirical tests according to research questions, but It's like the application of a method, and like the heterogeneity analysis in economics. In order to better meet the objectives of Land, please add relevant policies and suggestions to solve the real problems of housing market according to the research results.